# CBCT and Intra-Oral Scanner: The Advantages of 3D Technologies in Orthodontic Treatment

**DOI:** 10.3390/ijerph17249428

**Published:** 2020-12-16

**Authors:** Alessandra Impellizzeri, Martina Horodynski, Adriana De Stefano, Gaspare Palaia, Antonella Polimeni, Umberto Romeo, Elisabeth Guercio-Monaco, Gabriella Galluccio

**Affiliations:** Department of Oral and Maxillofacial Sciences, “Sapienza” University of Rome, via Caserta 6, 00161 Rome, Italy; martinahorodynski@gmail.com (M.H.); Adriana.destefano@uniroma1.it (A.D.S.); gaspare.palaia@uniroma1.it (G.P.); Antonella.polimeni@uniroma1.it (A.P.); umberto.romeo@uniroma1.it (U.R.); elisaguercio@hotmail.com (E.G.-M.); gabriella.galluccio@uniroma1.it (G.G.)

**Keywords:** digital technologies, dental movement, intraoral scanner, plaster cast, impacted canines, laser, orthodontics movement

## Abstract

Background: The aim is to demonstrate the validity of the monitoring through intraoral scanner of the dental movements and the real impact, advantages, and convenience, in terms of treatment time and efficiency gain, to frequently monitor a patient with the scanner application. Methods: A movement control of palatally impacted canines was performed, surgically treated with laser opercolectomy. Three-dimensional models of the patient’s dental arch were obtained with intraoral scanner during a monitoring time of 4 months. The STL (Standard Triangle Language) files were superimposed with the 3D models extrapolated from the pre-operative CT (Computerized Tomography). The measurements of eruption, exposed palatal and vestibular areas, and distances between the canines and the incisors were performed, using digital technologies and with a digital caliber. Results: Descriptive and inferential statistical analysis of the data obtained from both conventional and digital monitoring has been realized and performing the T Student Test for paired data. Conclusion: The advantages of digital monitoring are numerous, like the possibility to reduce the error of method caused by manual measurement on plaster casts and the possibility to compare the pattern and amount of eruption of the canine in the same patient overtime.

## 1. Introduction

Digital technologies are nowadays widely used in the several branches of dentistry. Literature reports that several digital methods have been applied in orthodontics, gradually modifying normal orthodontic practice in the last years. The popularity and availability of virtual technology in orthodontics for the replacement of hard-copy records with electronic records is growing rapidly, with a move towards a “digital” patient for diagnosis, treatment planning, monitoring of treatment progress, and results [1,2,3].

Digital scanning can be used in orthodontics for a variety of applications [4,5,6,7].

Making an accurate dental impression is one of the most essential and time-consuming procedures in dental practice. During this procedure, it is crucial to ensure the reproduction of the intraoral condition as accurately as possible.

The validity, reliability, and reproducibility of digital models obtained from an intraoral scanner allows to get dental measurements for orthodontic purposes [8].

The accuracy of a dental impression is determined by two factors: “trueness” and “precision.” Trueness is defined as the comparison between a reference dataset and a test dataset. Precision is defined as a comparison between various datasets obtained from the same object using the same scanner [9,10,11,12,13].

The aim of the study is to investigate the validity of the monitoring through intraoral scanner of the dental movements and the real impact, advantages, and convenience, in terms of treatment time and efficiency gain, to frequently monitor a patient with the scanner application. To establish validity of the new method, it was compared with the standard means widely used until now: conventional monitoring based on the study of plaster casts (gold standard).

## 2. Materials and Methods 

In this cohort study, a monitoring of dental movement of palatally impacted canines was performed, surgically treated with an innovative laser approach, avoiding the application of any type of orthodontic traction. 

The study was carried out on patients referred to the Orthodontics UOC of the Department of Odontostomatological and Maxillo-Facial Sciences of the “Sapienza” University of Rome. The period of recruitment of the patients was 3 months.

All the patients were informed about the content of the study, treatment methods, and potential risks and benefits, before providing written informed consent to take part in this study. The study received approval from the Ethical Committee of Sapienza University of Rome (#4389) and was registered in the international public register.

A preliminary investigation was performed to estimate the power of the study (PS) and to establish the effect size (ES) of the population sampled for the experimental study. It was calculated a statistical significance to determine an exact number of patients to become a study.

Suppose we want to estimate the prevalence of a disease (impacted canines) in a population. Through the study of the sample, we want an estimate of the prevalence with a certain precision and a chosen level of confidence. The size can be calculated, with a 95% confidence level, using the following Formula (1):(1)$$n=\frac{1.96^{2}\;P_{att\;}\left(1-P_{att}\right)}{D^{2}}$$
with *n* = Sample Size, *P_att_* = Prevalence estimating and *D*^2^= Absolute Precision

In our case with 95% confidence level and this data, when:

*P**_att_*: 96.5% (0.965)

*D*: 10% (0.1)

Therefore:

*n* = 1.96^2^ ∗ 0.965 ∗ (1 − 0.965)/0.1^2^ = 12.97

That is 13

If the sample size recommended was >5% compared to the population from which it is extracted, its sample size could be reduced; with statistical analysis, it was obtained that the sample size significance for this study was 13 canines.

The inclusion criteria considered in the experimental study design were:patients with palatally impacted canines;male and female;age between 12 and 25 years;patient will be reliable for follow-up;understands the protocol and can give informed consent.

The Exclusion Criteria were: Non-cooperative patients;inoperable patients;vestibular impacted canines;mandibular impacted canines;systemic pathologies;subject in drugs therapy.

The permanent maxillary canine is the most impacted tooth after third molars, followed by upper central incisor, lower second premolar and lower second molar [14,15,16]. It has a predilection for the palatal side with a tendency to the unilateral inclusion. To determinate the palatal inclusion of the canines, the patients were examined carefully: the palate was palpated to evaluate the presence of the bump due to the canine impaction and there was the absence of the canine draft in the fornix.

The final experimental sample was constituted of 10 patients, 5 females and 5 males; of these, 5 patients showed a bilateral inclusion and the other showed 5 a monolateral inclusion of the canine, for a total of 15 canines. Patient characteristics that were collected are age, sex, number, and location of the impacted canines. An orthopantomography (OPT) and a CBCT (Cone Beam Computed Tomography) were requested from patients at the beginning of therapy, to accurately evaluate the cases before surgery [17]. An evaluation of the prognosis of impacted canines was performed on OPT by two orthodontists [18], according to Ericson and Kurol [19].

The examination of the CT allowed to evaluate the three-dimensional morphology of the impacted tooth, its location and inclination in the three planes of space, the depth and the type of inclusion, and the relationships with the other elements (Figure 1).

To get an accurate initial situation, the Invesalius 3.1 software (CTI, Amarais, SP, Brazil) was used to export the CT in STL format. (Figure 2)

It is an open-source software for reconstruction of computed tomography and Magnetic Resonance Imaging [20]. The first step is to import CT files, then the range of values that corresponds to the radiopacity of the selected X-ray images that we want to be part of the three-dimensional model is chosen. At this point, a surface is generated, and it is obtained a 3D model in STL, which can be exported and meshed with the STL models of the intraoral scans.

The surgery was performed with an experimental laser disinclusion approach, using laser to make opercolectomy [21,22,23]. The treatments were carried out by the same operator.

After the surgery, in order to monitor the movement of the impacted dental elements, patients were checked on three control visits at one week, two months, and four months, in which intraoral photographs, conventional impression, and intraoral scans were carried out. Carestream 3500 intraoral scanner was used to take digital scans (accuracy: 30 µm), and Nikon D90 camera (Nikon Corporation, Tokyo, Japan) was used to take clinical photographs. Impressions and scans were made always by the same operator with the same procedures to standardize the data acquisition.

The three-dimensional models of the patients obtained with the scanner at 1 week, 2 months (T1), and 4 months (T2) were superimposed with the 3D models extrapolated from the pre-operative CT (T0), using the Meshlab program.

The measurements performed are:canine eruption at T1 and T2,maximum length of the vestibular surface at T1 and T2,maximum length of the palatal surface at T1 and T2,distance between canine and central incisor at T1 and T2,distance between canine and lateral incisor at T1 and T2.

It was possible to millimetrically evaluate these dimensions, and to calculate their changes over time precisely, always taking the same points of reference—i.e., the cusp of the canine and the most apical points of the palatal gingival margins of central and lateral incisor and of the canine itself—both on the palatal and the vestibular surface. The partial eruptions, obtained from T0 to T1 and from T1 to T2, were measured by superimposing the models and calculating the distances between the initial position of the cusp of the canine and its new position at T1 and T2. The maximum eruption of the tooth was measured by calculating the distance in mm between the cusp of the canine on the CT and the cusp at T2, on the superimposed models (Figure 3).

The position of the erupting tooth in respect to the central and lateral incisors was calculated, measuring the distance, always on superimposed models, of the T1 and T2 cusp with the palatal zenith of both the central incisor and the lateral incisor.

The superimposition of the model at T1 and T2 allowed to always consider the same point on the gingival margin, making the measurements accurate and repeatable (Figure 4).

The maximum lengths of the palatal and vestibular surfaces of the erupting teeth were measured at T1 and T2, overlapping the models and taking cusps and the most apical point of the vestibular and palatal gingival margin of the canine as reference points (Figure 5).

Then, the Image J program was used. It is an open-source image processing program, designed for scientific multidimensional images. It was possible to obtain the value of the palatal and vestibular areas of the erupting teeth at T1 and T2 by importing 2D images of the surfaces, created with MeshLab, on the program and inserting the maximum length values. The program measures the area of the surface contained in the perimeter, thanks to a pixel-mm conversion (Figure 6 and Figure 7).

At the same scheduled times, the conventional monitoring was performed. It consists of a clinical evaluation and on the analysis of the plaster casts, obtained by conventional impressions taken at each control visit. The measurements on the plaster casts were performed with a digital caliber (accuracy ± 0.03 mm). On the plaster casts, the same measurements that were also realized with digital monitoring allowed to obtain the values of:maximum length of the palatal and vestibular surface at T1 and T2,distance of the cusp from the zenith of central incisor and lateral incisor at T1 and T2,the eruption at T1 and T2 (the eruption can’t be calculated in a realistic way, but only by measuring the approximate height between the cusp and the palatal mucosa).

It was not possible to calculate the data relative to the areas because of the cited analogical techniques. The two monitoring methods were compared by evaluating the difference obtained for each measured data (eruption, distances between canine and incisors, and maximum length of vestibular and palatal surface). This comparison allowed to demonstrate the eventually present difference in precision between digital and conventional measurements.

A descriptive and inferential statistical analysis of each data obtained was realized from both conventional and digital monitoring, determining the statistical averages, the percentage increase, and performing the T Student Test for paired data.

All the data were measured twice by two observers to determinate the error of the methods, the correlations and the mean differences, and to compare the validity of the two monitoring methods.

## 3. Results

The data analysis of impacted canines performed on OPT reveals that 7 canines had a positive prognosis, while the remaining 8 were negative (Table 1).

Scheme 1 contains all the eruption data, calculated by superimposing the STL files through the Meshlab software, for canine in T0–T1 (corresponding to eruption from CT to 2 months), T1–T2 (from 2 to 4 months), and the total eruption movement T0–T2 (at 4 months).

The mean values of eruption, measured with digital and conventional methods, are reported in Table 2.

With the same method, the mean values of the distances (in mm) between the cusp of the canines and the apical point of the gingival margins of the central and lateral incisors at T1 (2 months) and T2 (4 months) were evaluated (Table 3).

Table 4 reports the mean values of maximum lengths of the vestibular and palatal surfaces of the canines at 2 months and 4 months.

The vestibular and palatal areas at T1 and T2 were calculated through the IMAGEJ program. Moreover, the differences between the size of the palatal and of the vestibular areas at T1 and T2, the percentage increase and the mean values were calculated (Table 5 and Table 6).

The T Student Test has been performed for the vestibular and palatal areas, comparing the values obtained at T1 and T2. The *p* values obtained are statistically significant (*p* = 0.00032 for the vestibular area, and *p* = 0.00001 for the palatal one). Starting from the eruption, the average speeds (in mm/day) in the interval T0–T1, T1–T2, and T0–T2 were calculated (Table 7).

Finally, the two types of monitoring were compared by evaluating the differences (in mm) between the measurements made on digital models and plaster casts (Table 8):the average difference of the total eruption is 1.09 mm,the average difference of the distance between the cusp and the incisor is 0.20 mm,the average difference of the length of the surfaces is 0.19 mm.

T test Student’s was applied for paired data of the total eruption, fixing the *p*-value < 0.05, and obtaining statistically significant *p* values (*p* = 3.25746 × 10^−8^). The interpretation of these results translates into a discrepancy between the two methods of measurement and a greater accuracy in monitoring with digital technologies compared to the conventional method.

All the data were measured by two different operators to determine the method error of digital and analogical measurements (Table 9, Table 10 and Table 11).

For each compared data, the correlation coefficient, the relative and absolute errors, and the mean differences were calculated (Table 12).

A coefficient of correlation close to 1 indicates a “positive” and “strong” relationship between the two examined variables. In other words, if one variable is increased, the other one will increase too. In this case in particular, it is observed a mean coefficient of correlation higher for digital measurements (0.9965) than the analogical ones performed with caliber (0.9665). This indicates a stronger correlation between the data of the digital method, suggesting less dependence on who makes the measurement itself, i.e., it is independent of the operator. This result is also supported by the absolute error values and the mean differences, which are both in favor of the digital method with mean values of 0.695% and 0.520% respectively, against 3.165% and 4.060% of the analogical one.

## 4. Discussion

Based on the results obtained, digital and conventional monitoring were compared. Digital technologies allow to obtain data that have a great clinical importance, because they make possible the appraisal of the value of the eruption of the impacted canines, speeds of eruption, amount of available vestibular and palatal areas, and possible approach to the central and lateral incisors.

From the results of the present study, the canines are spontaneously erupted on the palate over a period of 4 months, on average 4.49 mm. Of these, 2.71 mm occurred in the first 2 months.

Using the descripted exposure technique, which doesn’t comprehend the immediate positioning of orthodontic anchorage to realize the traction, it is possible to correctly evaluate the real spontaneous eruption of the canine.

The T test suggests that the difference in eruption and exposure of the palatal and vestibular areas in the two periods considered (T0–T1, T1–T2) are statistically significant data, being the *p*-value lower than 0.05. The exposed palatal surface is always bigger than the vestibular one. The patients with larger exposed area had mucosal impaction of the canines. The exposure of the areas is always greater in the first two months, with an increase of the available surface of 40% in the last 2 months.

As a further evaluation, the direction of eruption of the canine has been considered in relation to the possible approach to the central and lateral incisor, using the superimposition mode through the Meshlab software. It emerged that in one patient the canine approached the lateral incisor, in two cases it approached the central, and in three cases it approached both. It was verified that it is due to the inclination of the tooth, considering the unchanged inclination of the canine in all the superimpositions, from the initial one calculated from the CT image.

The speed of eruption of the impacted canines was also considered. The average speed in 0-T1 (0.058 mm/day) is greater than that in T1–T2 (0.030 mm/day). The total speed in 0-T2 is 0.037 mm/day. From this data, it can be deduced that the speed was greater in the first 2 months after the surgery and it tends to slow down afterwards.

Based on the significant differences, between the values of the measurements performed in the two types of monitoring and considering the statistically significant *p*-value obtained through student’s T test, it is possible to assert that monitoring with digital technologies is more precise than conventional monitoring, allowing to make affordable measurements. On the other hand, the analogical monitoring can’t provide realistic and repeatable measurements of dental movements, and neither the photographs and casts of the same patient taken at different times can be compared or overlapped [24,25].

Several authors demonstrated that the analogical procedures have errors derived from the distortion of elastomeric impressions, disproportionate water/powder ratio of dental plaster, or unsuitable storage conditions of physical impressions or gypsum casts [11,24].

With digital models fabricated from alginate impressions, moreover, fine details of tooth anatomy might be lost because of the limited ability of the impression material to flow into areas with undercuts, and potential shrinkage upon desiccation can compound the problem [22,26].

Additional loss of information may be related to the scanning process of the plaster casts because the accuracy of a digital model is limited by the resolution of the scanner [27].

Digital models represent the intraoral situation more accurately because there are fewer sources of error. It is logical to assume that when processing steps are eliminated in the production of digital models, the models will be more accurate [26].

Digital technologies make possible the evaluation of useful clinical data not obtainable with conventional monitoring: precise value of eruption of the impacted canines, speeds of eruption, amount of available vestibular and palatal areas, and possible approach to the central and lateral incisors [27].

Regarding the surgical-orthodontic canine’s disinclusion protocol used in this study, it differs from the conventional approach that contemplate an immediate positioning of the bracket or button to exert the orthodontic force. This procedure can be difficult due to the possible bleeding or to the reduced area of surgical exposure [28].

The level of exposure of the canine’s crown at 4 months, observed in this study, can be considered enough for the positioning of an orthodontic anchorage suitable for the subsequent traction. Therefore, it allowed to easily start the following phase of the fixed orthodontic therapy to bring the canine to its physiological position in the dental arch [29].

The acquaintance of this data allows to know when to start fixed therapy, which is a great clinical advantage, especially for patients who, for personal reasons, cannot be frequently evaluated, or in non-compliant patients or suffering from systemic diseases. It also makes work time organization easier and more efficient. The cases in which there is not a certain pattern of correct eruption and the cases in which there is an excessive proximity to the near structures are also treated with greater predictability.

Several authors [30,31] in their study performed a conventional monitoring of impacted canine’s movement after surgery exposure of the crown and the appliance of traction force on it. The movement’s control consists on the clinical evaluation of the eruption at each check-up until the end of fixed therapy. The duration of treatment and the number of visits performed have been recorded [32].

Kokich et al. [33] differs from other authors for having applied the same type of monitoring on surgically exposed canines without application of means of traction.

Kocsis et al. [34] used orthodontics screw as mean of traction for impacted canines. They performed a clinical monitoring making an intraoral examination of patients at 4 week-interval control visits. Anterior occlusal radiographs and periapical X-rays were taken at each visit.

In literature, there are no methods for monitoring the eruption of palatally impacted canines after surgery that contemplate digital measurements.

The limitations of the study are the restricted sample of monitored patients, and the error due to the devices used for monitoring: CBCT, conversion programs, CS 3500, and other software. Another disadvantage is the cost of the intraoral scanner, but it is conceivable that it can be amortized over time.

## 5. Conclusions

Digital monitoring allows to reduce the approximations caused by manual measurement with digital caliber or ruler and compass on plaster casts, and to make assessments and comparisons over time of the same patient.

The main advantages of digital monitoring are the possibility of making more precise measurements of distances, and the reduction of work time, due to the abolishing of the need to request plaster casts to the dental technician, which also means the reduction of the costs for both the laboratory and the patients. By eliminating the steps of the analogue impressions and of the plaster casts, the details are certainly represented with more precision and accuracy and there is a minimum error accumulated. Moreover, this digital workflow can be managed entirely by a single person, which represents a further saving of time.

The clinical advantage lies in the less discomfort of the patient who does not tolerate the classic impressions in alginate.

The application of the digital technologies in the monitoring helps the orthodontist to make clinical decisions supported on measurable data and not just on clinical experience.

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
