# Peer review of "CBCT and Intra-Oral Scanner: The Advantages of 3D Technologies in Orthodontic Treatment"

_ijerph, 2020, doi:10.3390/ijerph17249428_

Round 1
Reviewer 1 Report
Dear authors,
Dear Authors,
Congratulations on your work!
In general, the work is well explained.
However, in my opinion the results should not be so linearly extrapolated since you did not take into account aspects such as:
- The difference in available space for the spontaneous eruption of canines between the different cases. It is certain that you considered the slope and the distance to the incisors, but not the total area available in the arch and it could influence the results.
From what I would like to see this aspect explained or the reasons why it was not considered in the study.
Best regards,
Author Response
Dear Reviewer,
Thanks for your review and for the best wish for our work! With pleasurewe response to your curiosity.
We dont' have included in our study the space availability in the arch for
the canines, because this parameter it's one of the causes for the nature
of impacted canines. So, we should have included this in the preliminary
study, but we have considered other different parameters as inclination and
distance to the incisors.
The availability of the space in the arch, in our opinion, does not find
correlation and does not affect the eruption movement of the canine on the
palate, after the laser exposure. However, the absence or presence of space in the arch could influence the
subsequent phase of approach and alignment of the impacted tooth,
and so it should be considered in further more in-depth studies.
Best regards.
Reviewer 2 Report
1) The affiliation of the authors is not correctly described.
2) In the inclusion criteria, how was it determined that the canines were impacted by palatine?
3) It is necessary to review the title of table 8 of the manuscript.
4) In tables 8, 9, 10 and 11 it is necessary to include images with higher resolution.
5) Bibliographic references should not be described in the conclusions.
6) Bibliographic references are not written according to the journal's standards.
7) In the manuscript there are paragraphs without bibliographical reference.
8) In discussion it is necessary to describe the limitations of the study.
9) The conclusions are very extensive.
Author Response
1) The affiliation of the authors is not correctly described.
Response 1: The affiliation of the authors has been corrected.
2) In the inclusion criteria, how was it determined that the canines were impacted by palatine?
Response 2: A paragraph has been added with the explanation in the materials and methods section. (110-114)
3) It is necessary to review the title of table 8 of the manuscript.
Response 3: The title of the table 8 has been changed.
4) In tables 8, 9, 10 and 11 it is necessary to include images with higher resolution.
Response 4: Images with higher resolution have been loaded (Table 8,9,10 and 11)
5) Bibliographic references should not be described in the conclusions
Response 5: The references in the conclusions have been eliminated.
6) Bibliographic references are not written according to the journal's standards.
Response 6: Bibliographic references have been rewritten according to the journal's standards.
7) In the manuscript there are paragraphs without bibliographical reference.
Response 7: The bibliographical references have been added to the paragraphs who did not have them.
8) In discussion it is necessary to describe the limitations of the study.
Response 8: The limitations of the study have been added in the discussion part. (358-361)
9) The conclusions are very extensive.
Response 9: The conclusion have been reduced.
Round 2
Reviewer 1 Report
Dear authors,
I am enlightened the justification given.
Best Regards,
Reviewer 2 Report
Authors should check the bibliographical references.
In the modified document, the bibliographical references are not described according to the journal's regulations.